# Fetal Middle Cerebral Artery Pulsatility Index in No-Risk Pregnancies: Effects of Auditory Stimulation and Pregnancy Order

**DOI:** 10.3390/ijms21113855

**Published:** 2020-05-29

**Authors:** Ljiljana Jeličić, Svetlana Janković, Mirjana Sovilj, Tatjana Adamović, Ivana Bogavac, Aleksandar Gavrilović, Miško Subotić

**Affiliations:** 1Cognitive Neuroscience Department Research and Development Institute “Life Activities Advancement Center”, Belgrade 11000, Serbia; tadus3@gmail.com (T.A.); ivbogavac@gmail.com (I.B.); m.subotic@add-for-life.com (M.S.); 2Department of Speech, Language and Hearing Sciences, Institute for Experimental Phonetics and Speech Pathology, Belgrade 11000, Serbia; iefpgmir@gmail.com; 3Department of Acute Perinatal Pathology, Belgrade University Medical School, Belgrade 11000, Serbia; svetlanajankovic.r@gmail.com; 4Clinic for Gynecology and Obstetrics Narodni front Belgrade, Belgrade 11000, Serbia; 5Department of Neurology, Faculty of Medical Sciences, University of Kragujevac, Kragujevac 34000, Serbia; agavrilovic@hotmail.rs; 6Department of Neurophysiology, Clinic of Neurology—Clinical Center “Kragujevac”, Kragujevac 34000, Serbia

**Keywords:** Pulsatility index, fetal auditory perception, parous and nulliparous pregnancy, middle cerebral artery

## Abstract

Pulsatility index (PI) values in a fetal middle cerebral artery (MCA) were compared in no-risk pregnancies to examine the differences related to auditory stimulation test and pregnancy order. The study included 196 women with no-risk pregnancies selected from the database of more than 1000 pregnant women divided into two groups. Group 1 consisted of 98 nulliparous women (C1 = 98) and Group 2 consisted of 98 parous women (C2 = 98). All pregnant women were of comparable age and fetal gestational age (GA) when MCA-PI values were recorded. Measurements of PI values in fetal MCA were obtained before and immediately after the application of fetal auditory stimulation test. The MCA-PI measuring was conducted in the period between the 36th and the 41st week of GA. The results showed that PI baseline values and PI values after defined auditory stimulation were significantly different when measured in nulliparous women compared to parous women (*p* = 0.001; *p* = 0.003, respectively), while no group differences were observed in relative PI value changes due to auditory stimulation. These findings suggest that hemodynamic changes in fetal MCA caused by defined auditory stimulation measured by PI value changes may be valuable in the assessment of fetal auditory perception functionality and its development.

## 1. Introduction

The investigation of fetal behavior and its reactions to different stimuli indicates the cortical structure’s development and maturity. Observing and examining the maturity of the fetal auditory system is crucial, as it may indicate the auditory potentials necessary in speech and language development. Moreover, it may point to a need for the application of stimulation procedures. Certain follow-up studies, including those from our own research group, have confirmed the association between fetal reactions to defined sound stimulus, neonatal outcomes, language development, and early child development [1,2,3]. Furthermore, there is evidence that the intrauterine auditory stimulation may have an important role in shaping later auditory development [4], which may reflect not only on speech-language development, but also on the overall child development. These findings point to the importance of auditory perception estimation at the early ontogenetic age.

Different methodologies applied in this research field have confirmed fetal auditory system maturity and functionality [5,6,7,8,9,10,11,12,13,14,15,16,17]. The examination of fetal auditory perception by acoustic stimulation may point to auditory pathways maturity and capability of fetal auditory perception. In that manner, the method of Prenatal Hearing Screening (PHS) was developed and applied by Sovilj and Ljubic in 1992 [18], while its application in further studies gave reliable findings with regards to fetal auditory reactions [2,19,20,21], revealing fetal maturity as well as presenting a potential for a reliable prenatal hearing screening test. The PHS method is based on measuring the Pulsatility Index (PI) before and after defined auditory stimulation [19]. Research in this field has shown that PI is the most appropriate parameter in examining fetal auditory reactions caused by defined sound stimuli when compared to Resistance index and bulbar movements [19]. 

Reference ranges for PI in the middle cerebral artery (MCA-PI) in normal and complicated pregnancies have been established in many studies [22,23,24,25,26,27,28,29]. When observing the reference range in normal pregnancies, there is always the parabolic shape of the PI curve which indicates the decrease of fetal MCA-PI with increasing gestational age (GA) [24,25]. The PHS application has the aim to establish differential and diagnostic parameters of fetal auditory reactions to pregnancy conditions which would be indicators for fetal well-being estimation, as well as early stimulation procedures’ application. The fetal auditory system becomes functional in the 25th GA [13]: the fetus begins with active hearing when the ear and cochlea are structurally formed and when hearing becomes the main information channel [30]. Research in the field of prenatal auditory perception indicates that auditory nerve pathway is formed by increasing the response rate with increasing GA; this means that due to vibroacoustic stimulation, in 7% of fetuses, responses are present between the 20th and 22nd GA, in 22% of fetuses, responses are present between the 24th and 26th GA, in 89% of fetuses, responses are present between the 26th and 28th GA while in 100% of fetuses, responses are present between the 30th and 32nd GA [31]. A recent study also reports that fetal movement responses to vibroacoustic stimulation are consistent after the 28th GA, suggesting that fetal hearing develops at or before the 28th week of intrauterine life [32]. Even in studies that indicate an incomplete maturation of fetal auditory system, it is emphasized that the developing auditory system enables response to sound in utero [33], which is shown by studies that measure fetal movement or heart rate response to sound at around the 25th and 27th GA [34]. This confirms that hearing has improved dramatically by term birth [35]. On the other hand, there has been growing evidence which proves that the development and maturation of the auditory system depend substantially on the afferent activity supplying inputs to the developing centers [36]. Taking into account all these physiological aspects, changes of PI in the middle cerebral artery (MCA) may be detected after the 27th gestational week as an indicator of functional and psychological maturity in the fetal auditory system. 

Considering the psychological and physiological differences between the first and the second pregnancy, our aim was to establish whether these can be observed at the level of fetal auditory perception. As it is known from the literature, MCA-PI is higher in parous than in nulliparous women [29,37], but we assume that these differences do not affect fetal auditory reactivity which is influenced by fetal auditory system maturation. In that manner, this study aims to compare Pulsatility index (PI) values in the fetal middle cerebral artery (MCA) and examine the differences related to auditory stimulation test in nulliparous and parous women with no risk pregnancies.

## 2. Results

The study included 196 healthy pregnant women, all singleton pregnancies with no known risk or complications. The sample was divided into two groups: C1 (C1= 98 nulliparous women) and C2 (C2 = 98 parous women). Nulliparous women from group C1 were matched 1:1 for chronological age, and gestational age with parous women from group C2. Maternal age in both groups ranged between 25 and 34 years, with the mean age of 30 years. The measurement of MCA-PI before and after defined sound stimulation was obtained during the period from the 36th to 41st GA. The mean gestational age in weeks was 38.9, as shown in Table 1.

Figure 1 shows the PI baseline in MCA (PIB) in groups C1 and C2. The analysis of PIB values before the application of defined sound stimulation showed that these values are within the normal reference range of PI values reported in the literature [27].

The mean PIB value (1.48) in parous women (the C2) group is higher than the mean PIB value (1.34) in nulliparous women (the C1) group (Table 1). The comparison of PIB in C1 and C2 by One Way Analysis of Variance (ANOVA) reveals a statistically significant difference (*p* = 0.001) between these groups (Table 1).

The PI values following auditory stimulation—PI reactions (PIR) in C2 are on average also higher compared to C1 (Table 1). One Way ANOVA analysis points to the statistically significant difference (*p* = 0.003) between the groups (Table 1). 

Based on the findings that PIR depends on pregnancy order, we introduce two new variables: relative PI change (RePI) and absolute PI change (AbsPI).

The analysis of Absolute RePI changes (AbsPI) and Relative PI changes (RePI) by One Way ANOVA, as parameters that measure fetal reactivity to auditory stimulation, showed that there was no statistically significant difference between groups C1 and C2 (Table 1).

When observing RePI as a parameter of fetal reactivity to the defined auditory stimulus, it is noticed that RePI values in C1 and C2 may have positive and negative values. Namely, measured PI values after auditory stimulation may be higher or lower than the baseline PI values. In both groups, PIB values were approximately twice as often higher than PIR values. If groups C1 and C2 are divided into two subgroups: C1 = C1− and C1+; C2 = C2− and C2+, where the group in which RePI < 0 is marked with “−”, and the group in which RePI > 0 is marked with “+”, then in the group C1 29% of pregnant women belong to “−“ (C1−) and 71% belong to “+” (C1+) subgroup. In the group C2 35% of pregnant women belong to “−“ (C2−) and 65% belong to “+” (C2+) subgroup.

Bivariate correlation analysis of PIR values and RePI values shows that there is a correlation between these two parameters (Pearson coefficient of correlation r = −0.646, *p* = 0.000). Considering Cohen’s definition (1992) which says that effect size for R^2^ = 0.01 is a small, R^2^ = 0.09 is a medium, and R^2^ = 0.25 is a large effect, our results point to the large effect size as calculated R^2^ is 0.42.

The result of linear regression analysis of the dependence of RePI on PIR is given in Figure 2. It can be seen that as the PIR value increases, the RePI value decreases.

Assuming that the AbsPI is the magnitude of the fetal response to the auditory stimulus regardless of the direction of the reaction (positive or negative), univariate two way ANOVA of the two-factor AbsPI dependence (group: nulliparous/parous and direction of the fetal response: positive-RePI > 0/negative-RePI < 0) indicates that the direction of the reaction has an effect on AbsPI (df = 1, F = 8.25, *p* = 0.005) while Group has no effect as expected (Table 1). Furthermore, there is no interaction between Group and the direction of the reaction (df = 1, F = 2.97, *p* = 0.086).

Bivariate correlation analysis of PIR and AbsPI values in the four subgroups (Table 2) shows that there is a correlation between these two parameters, with medium to large effect sizes.

Unlike the correlation coefficient between RePI and PIR which has a negative value, the correlation coefficients between AbsPI and PIR in C1− and C2− have a positive sign, while in the C1+ and C2+ group they have a negative sign. In the first case (“−” subgroups), with the increase of PIR value, the value of AbsPI increases as well. On the other hand, in the second case (“+” subgroups) with the increase of PIR value, the value of AbsPI decreases.

Figure 3 presents the result of linear regression analysis of the dependence of mean AbsPI values per GA, showing an increasing trend of mean AbsPI values by increasing GA. This may point to the relationship between fetal reactions and the maturation of cortex and auditory pathways as well. However, further research is necessary before any strong conclusion can be made.

## 3. Discussion

The focus of our study was to examine to what extent PI changes in MCA, caused by the application of PHS method, may be associated with fetal response to auditory stimulation. The application of PHS method in some other studies obtained reliable findings with regards to fetal auditory reaction, where the aim was to establish a reliable hearing screening test [19,20,21]. Our approach is based on the fact that PI changes could be associated with auditory stimulation. Thus, we aimed to determine whether these PI changes do or do not depend on PI baseline. In that case, PI value changes caused by defined auditory stimulation may be a candidate for auditory perception examination. 

MCA PI changes were analyzed in relation to pregnancy order (i.e., in parous and nulliparous women). The PHS was applied between the 36th and 41st GA as this is the period within which the fetal auditory system has reached sufficient functional and psychological maturity [38]. Our findings related to measured PI baseline (PIB) in parous and nulliparous women with no risk pregnancies reveal that these values are compatible with reference ranges of PI values established in other research [22,27]. The obtained results also point to significantly higher PIB values measured in parous women compared to nulliparous women which is in accordance with findings regarding the influence of parity on fetal hemodynamics [37]. It is also noticed that the PI values following auditory stimulation, PI reactions (PIR), are also significantly higher in parous women compared to nulliparous women. In the case of both parameters, it is shown that pregnancy order has an effect and that there is a statistically significant difference in PIB and PIR values between nulliparous and parous women. Since PIR is dependent on pregnancy order, it is not a suitable parameter for assessing the fetal response to an acoustic stimulus.

Considering the fact that all examined children were tested after the birth by neonatal hearing screening test, TEOAE, and all of them passed this test on both ears, the difference between PIR values in C1 and C2 group may not be linked only to fetal auditory perception. 

In order to minimize the effect of PIR dependence on PIB, parameters of RePI and AbsPI are also considered in the paper. The analysis of RePI and AbsPI changes which point to relative fetal cerebral circulation changes in MCA caused by a defined sound stimulus, shows that fetuses from parous and nulliparous women do not differ with regards to these changes. Such results are expected since all our participants in the groups C1 and C2 had pregnancies without any complications and their children passed TEOAE test during neonatal hearing screening procedure following birth and were recorded to have no hearing loss in the early postnatal developmental period. Therefore, the proposed parameters AbsPI and RePI may be used as candidates in the examination of the fetal reactions to sound stimulus.

Our analysis of the proposed parameter RePI shows there are two types of fetal reactivity: 

Type 1—a decrease of PI value following defined sound stimulation which indicates a decrease in MCA resistance, immediately after the stimulation and

Type 2—an increase of PI value following defined sound stimulation which indicates an increase in MCA resistance, immediately after the stimulation.

Our findings reveal a negative correlation between PIR and RePI values. As the fetal response to the auditory stimulus increases, the difference between PIB and PIR decreases. However, this could lead to the misleading interpretation of the findings, since the value of RePI can be both positive and negative. By solving the equation that represents a linear approximation of the dependence of RePI on PIR, we find that the interpolation line intersects the *x*-axis (PIR) at the value 0 = −0.5135 × PIR + 0.7171 which leads to PIR = 1.4. For PIR values that are less than 1.4, the difference between PIB and PIR decreases as PIR value increases. However, for PIR values that are greater than 1.4, the difference between the PIB and PIR values grows, but the sign of this difference is negative.

If we look at the formula for calculating RePI
RePI = (PIB − PIR)/PIB(1)
where: RePI—relative PI changes, PIB—PI baseline in middle cerebral artery (MCA), and PIR—PI reactions immediately after defined auditory stimulation in MCA.

It can be rewritten in the form
RePI = 1 − PIR/PIB.(2)

Bearing in mind that PI is calculated as [39]:PI = (v_s_ − v_d_)/v_mean_(3)
where: v_s_—systolic velocity, v_d_—diastolic velocity, and v_mean_—mean velocity during cardiac cycle.

Equation for the mean velocity during cardiac cycle is
(4)$$V_{mean}=\frac{\int_{0}^{T}v\left(t\right)dt\;}{T}$$
where T is the duration of cardiac cycle.

Now, the equation for RePI (Equation (2)) can be rewritten in the form:(5)$$RePI=1-\frac{\left(V_{sr}-V_{dr}\right)}{\left(V_{sb}-V_{db}\right)}\times \frac{\int_{0}^{Tb}v_{b}\left(t\right)dt}{\int_{0}^{Tr}v_{r}\left(t\right)dt}\times \frac{T_{r}}{T_{b}}$$
where parameters with subscript r denote values measured after the acoustic stimulation and parameters with subscript b denote the baseline values.

For the simplicity, the formula for RePI (Equation (5)) can be rewritten in form:(6)$$RePI=1-f\left(v\right)\times \frac{T_{r}}{T_{b}}$$
where: f(v) is term that depends on the velocity in fetal cerebri media, Tr is the duration of cardiac cycle after acoustic stimulation, and Tb is the baseline duration of cardiac cycle.

We can see that the value of RePI (Equation (6)) depends on the velocity and heart rate (duration of the cardiac cycle) of the fetus. For the moment, we can focus our attention on the term in the equation dependent on heart cycle duration. Suppose that the velocity and the heart rate can change independently, and the velocity-dependent term of the equation is constant. If the heart cycle after stimulation is higher than the baseline heart cycle duration, the term Tr/Tb is larger than one. In that case, the value of RePI may become less than one. When the heart cycle duration, after stimulation, is lower than the baseline hart cycle duration, then the value of RePI may be larger than one.

According to our results, the presence of the first and the second type of fetal reaction is similarly distributed in both groups. It may be assumed that these two types of fetal psychophysiological reactivity may be postulated at the genetic level. Dissociation of fetal reactivity is described in previous research, independent of ours [40], where it was observed that vibroacoustic stimulation can cause generalized activation or calming effect on the fetus. The authors Graham and Sokolov [41] observed and defined the activation of two different categorical response systems: defense and orientation. The defensive response reflects the generalized activity of organisms characterized by heart rate acceleration and body movement. The orientation reflects the calming effect of the body, characterized by slowing heart rate, which can be assumed to represent the excitation of attention mechanisms and perhaps even the evidence with regards to fetal cognition. This demonstrated activation of the defense or orientation system in utero, suggesting that the roots of sensory and motor behavior organization are found in the prenatal period [40].

Bearing in mind that the value of PI is also dependent on heart rate, obtained results may be interpreted as:

In PHS, type 1 of the fetal reactivity, which indicates a decrease in MCA resistance immediately after the stimulation, may be classified as a categorical defense system which generates generalized activity of the organism and may be related to the accelerated heart rate. Similarly, type 2 of the fetal reactivity, which indicates an increase in MCA resistance immediately after the stimulation, may be classified as categorical orientation system which reflects calming of an organism and may be related to slowing the heart rate, and likely contains some elements of attention. Further longitudinal research may provide indicators on whether these two types of fetal reactivity could be related to personality typology in later ages. Moreover, it should be noted that a similar analysis can also be conducted for the contribution of the term in equation for RePI that depends on the blood flow velocity in the cerebri media. In order to determine the influence of individual terms on the values of the RePI and AbsPI parameters, additional research is needed.

In general, the obtained results show that relative MCA-PI changes, caused by defined auditory stimulation do not depend on pregnancy order which is one of the prerequisites for using this procedure as a test for a prenatal hearing screening. However, as RePI values may be positive and negative, this result may be debatable. Therefore, the parameter of AbsPI is introduced because it represents the absolute RePI value and indicates the intensity of the fetal reactivity to auditory stimulation regardless of the reaction type (decrease or increase of PI after auditory stimulation). Furthermore, it is shown that this parameter does not depend on the pregnancy order.

It is also shown that the correlation coefficients between AbsPI and PIR in “−“ and “+” groups have different signs. One possible reason is the different contribution of blood velocity in cerebri media and heart rate to the value of the PI after acoustic stimulation.

In the observed sample, the increasing dependence can be seen between AbsPI and GA. Although this trend may be related to the known facts about fetal auditory pathways maturation, the results should be interpreted with caution. As the number of pregnant women per GA is not the same, obtained average AbsPI values for different GA do not have the same weight. Our analyses of the observed sample also show that there is no statistical difference between average AbsPI values for different GA. Consequently, further research in which the frequency of pregnant women with regards to GA is more precisely balanced will provide better understanding of the behavior of this parameter and give at least a tentative answer to the question of whether there exists a relationship between AbsPI and fetal auditory system maturation. However, this may be explained by the fact which states that continuous auditory pathways maturation occurs in the last trimester of pregnancy. Hearing threshold, as an important parameter which affects auditory perception, is around 40 dB in the period from the 27th to 29th GA, while it decreases almost to 13.5 dB in the 42nd GA [42]. 

## 4. Materials and Methods 

This cross-sectional study was a part of a larger experimental study conducted under the supervision of the Ministry of education and health of the Republic of Serbia during the period from 2011 until 2019. The participating institutions included the Clinic for gynecology and obstetrics—Clinical Center of Serbia, the Clinic for gynecology and obstetrics “Narodni Front” in Belgrade, the Institute for Experimental Phonetics and Speech Pathology in Belgrade and the Research and Development Institute “Life Activities Advancement Center” in Belgrade, Serbia. Our study included 196 women with no risk pregnancies, selected from a cohort of more than 1000 pregnant women, who took part in the longitudinal investigation of fetal auditory reactions in previous studies [2,20,21] and their relation to the neonatal and early child development [2,3]. Participating women were recruited from two state-funded clinics in the Serbian capital, Belgrade. The number of births in both clinics is around 13,000 per year, with pregnant women coming from all over Serbia as well neighboring countries, Montenegro, and Republic of Srpska. While we did not collect details of participants’ socioeconomic status or place of residence (only their medical data and gynecologist reports were consulted), we believe it is reasonable to conclude that the sample is representative of all pregnant women in the Republic of Serbia.

The anamnestic data was collected from the pregnant women’s medical records. The inclusion criteria for this study were: normal singleton pregnancy without complications of any kind, delivering a phenotypically normal live birth; accurate GA based on the last menstruation date adapted with ultrasound parameters, GA between the 36th and 41st week, normal fetal growth (between the 10th and 90th percentiles), and normal Doppler pattern of MCA and uterine artery (UA) arteries. The exclusion criteria were as follows: multiple pregnancies, history of hypertension, preeclampsia, eclampsia, diabetes, autoimmune conditions or any other medical condition, presence of any general chronic disease, biophysical profile which is less than 6 and fetal abnormalities. The participating pregnant women used no alcoholic beverages, tobacco, or any other type of psychoactive substances. The complete study protocol had been approved by the Ethical Boards of the Clinic for Gynecology and Obstetrics “Narodni Front” and the Clinic for Gynecology and Obstetrics—Clinical Center of Serbia (Project OI178027, Date: 9.February.2012., No 1/12), in Belgrade, which operates in accordance with the Ethical principles in medical research involving human subjects, established by the Declaration of Helsinki 2013. Written informed consent was obtained from all participants. 

The study involved a matched-control design. The original sample of over 1000 pregnant women included a smaller number of parous women than nulliparous women, the group of parous women who met the inclusion and the exclusion criteria was formed first (group C2, *n* = 98). An equal number of nulliparous women (group C1, *n* = 98) matched 1:1 on chronological age and GA to women from C2 group was randomly selected. The choice of matching variables, on age and GA, was based on the literature suggesting that the MCA-PI depends on the gestational week [22,23,24,25,26,27,28,29], as well as maternal age [28]. 

The study employed the method of prenatal hearing screening (PHS) [18,19,43] was Measurements of PI values in fetal MCA were obtained before (baseline) and immediately after the application of PHS, which is based on auditory stimulation. The MCA-PI measuring was carried out within the period from the 36th to 41st week of GA. GA was determined in relation to the last menstruation date based on ultrasound parameters. As the aim of our study was to compare PI values in low-risk pregnancies in order to examine differences related to fetal auditory reactions and pregnancy order, we selected the period between the 36th and 41st GA, which is when the fetal auditory system has reached sufficient functional and psychological maturity. The screening method for the auditory stimulation (PHS) was applied only once per pregnant woman. 

As it was already stated, PHS method is based on Doppler waves analysis which is performed by measuring the PI value. PI is the Doppler index which shows the ratio of absolute velocities within one Doppler waveform and does not depend on the angle of insonation (the angle between the ultrasonic wave and the direction of flow). This is why PI is often used in measuring the characteristics of blood flow in fetal and uteroplacental arteries. As the PI value predominantly depends on the blood flow velocity at the end of diastole, it is considered an indicator of the peripheral vascular resistance extent [44]. MCA-PI decreases with increasing GA after reaching a peak at around the 32nd and 34th GA [28]. Based on these facts, measuring measurements of PI value, either in fetal MCA or in the uterine artery (UA), may be a reliable assessment tool for fetal well-being estimation.

In the PHS procedure, on the basis of PIB values (PI baseline) and PIR values (PI values following auditory stimulation—PI reactions), we calculated the relative changes of PI values (RePI) which may point to changes in cerebral flow circulation due to auditory stimulation. In other words, we examine to what extent the PI changes in MCA caused by defined sound stimulus may be associated with fetal response to auditory stimulation.

Relative PI changes (RePI) (Equation (1)) and absolute PI changes (AbsPI) are calculated as:AbsPI = Abs(RePI)(7)
where: Abs is a mathematical operation which result is a positive value of equation, PIB is PI baseline, PIR is PI measured immediately after defined sound stimulation.

Relative change of PI values may be perceived as an indicator of the fetal auditory system response to defined auditory stimulation. All measurements of PI values changes were obtained in the absence of fetal movements and fetal breathing, which is defined by the absence of visible changes in more than five consecutive Doppler waveforms. Observation and measurement of fetal cerebral cortical circulation changes were performed on the ultrasound apparatus TOSHIBA, with Doppler and color Doppler features. Convex and sector probe with a frequency of 3.5 MHz was used. Cerebral circulation changes were measured in MCA which was visualized, and satisfactory flow velocity waveforms were obtained successfully in the first third of MCA with regards to Circle of Willis.

The defined sound stimulus is a digitally derived sound with the intensity of 90dB, frequency range of 1500–4500 Hz, and a duration of 0.2 s (click). It is applied by MIMS-GENERATOR SOUND STIMULANT (production by Inkomark, Belgrade, Serbia; Patent No. P 2010/0519) developed at the Institute for experimental phonetics and speech pathology in Belgrade. The sound stimulus was presented to each fetus only once in order to exclude the fetal habituation to stimulus, confirmed in previous studies [18]. In order to exclude the influence of a defined sound stimulus over the mother’s auditory system, each pregnant woman’s ears were covered by ear shells type EP-107. After birth, all children were examined by hearing screening test, TEOAE, which is a screening tool for determining if the cochlea is responding to sound in a typical fashion [45]. TEOAE has clinical importance as it is a non-invasive test for hearing disorders in newborn babies and in our study, it was applied in the 2nd and 3rd day following their births. All of the examined children passed the TEOAE test in both ears. 

Statistical analyses were performed using Statistical Package for the Social Sciences (SPSS) for Windows (v. 20.0). Each scale variable (PIB, PIR, RePI, AbsPI, GA, MA) was normally distributed. Therefore, parametric statistics are used and *p* values of less than 0.05 were considered significant.

Analysis of variance (ANOVA) was used for the comparison of group means. Univariate two way ANOVA (type III sum of squares in model) was used to analyze influence of the two-factors (Group: nulliparous/parous and direction of fetal response: positive-RePI > 0/negative-RePI < 0) on mean value of AbsPI. Bivariate correlation was used to measure the association between RePI and AbsPI values with PIR in groups C1 and C2. A full linear regression was applied to test the possible predictive effect of the model.

## 5. Conclusions

In this study, we show that PI value changes may be considered as complex measurement values that enable the examination of fetal auditory perception. This may be linked with fetal heart rate, vasoconstriction, or vasodilatation, but in this paper, our aim was to establish the link between RePI and AbsPI with fetal responses caused by auditory stimulation. As it was determined that changes of the RePI and AbsPI do not depend on pregnancy order, they may be used as the parameters of fetal auditory responses examination which should be subject to further research.

## Figures and Tables

**Figure 1 ijms-21-03855-f001:**
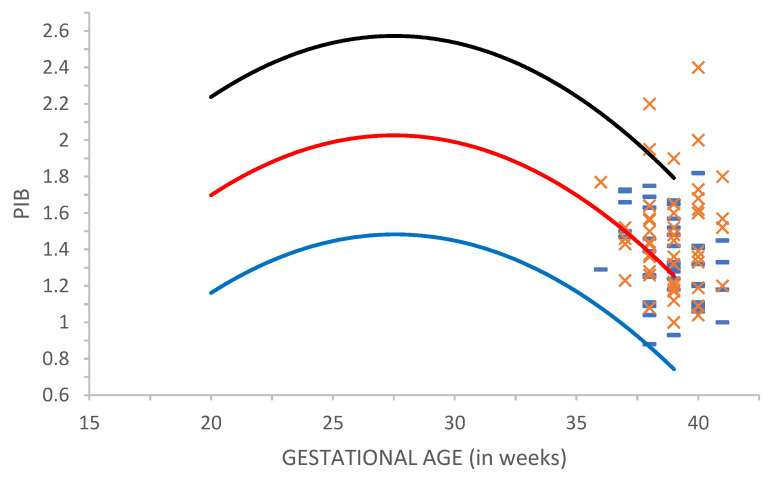
The values of Pulsatility index baseline in C1(**-**) and C2 (x) compared to referent values in Tarzamni [27]. The blue line represents the 10th percentile, the red line represents 50th and the black line represents the 90th percentile.

**Figure 2 ijms-21-03855-f002:**
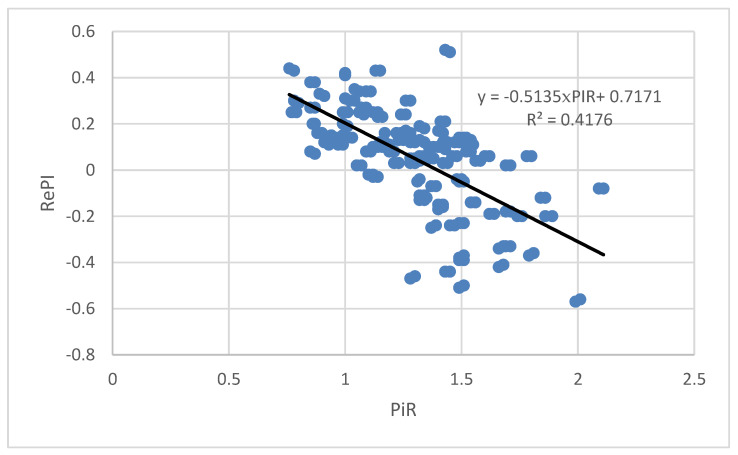
Dependence of RePI from PIR. Black line represents linear approximation.

**Figure 3 ijms-21-03855-f003:**
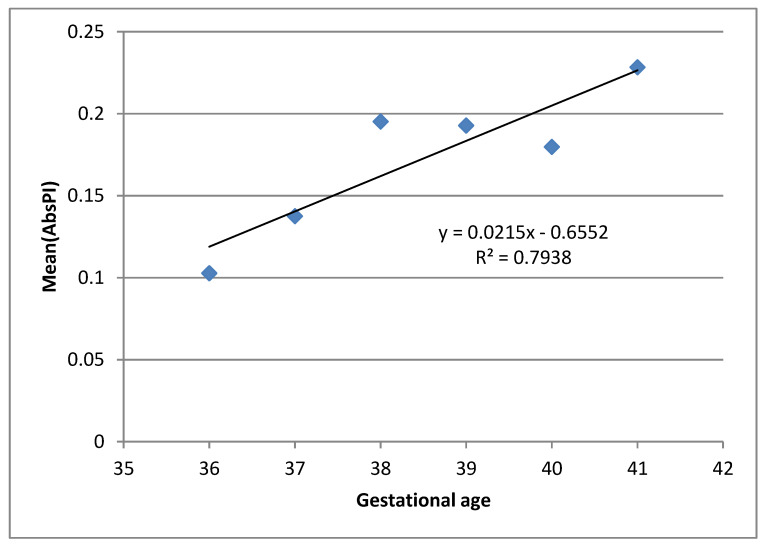
Dependence of mean AbsPI value up to gestational age (GA). AbsPI = absolute PI changes. Mean (AbsPI) represents the mean of all pregnant women AbsPI values in a given gestational week.

**Table 1 ijms-21-03855-t001:** Comparative analysis of maternal age, gestation week, absolute PI (AbsPI) changes, and relative PI (RePI) changes in groups C1 and C2.

Maternal Age, Gestational Age, and Comparative Analysis of PI Value Changes in Cl and C2 Group		Number (*n*)	Mean	Standard Deviation	Minimum	Maximum	Degrees of Freedom (df)	F Value	Significance (*p*)
Maternal age (years)	Cl	98	30	2.53	25	34	1	0	1
	C2	98	30	2.53	25	34			
Gestational age (weeks)	Cl	98	38.9	1.2	36	41	1	0	1
	C2	98	38.9	1.2	36	41			
PIB	Cl	98	1.34	0.23	0.87	1.83	1	11.158	0.001
	C2	98	1.48	0.32	0.99	2.97			
PIR	Cl	98	1.25	0.24	0.77	1.71	1	8.768	0.003
	C2	98	1.37	0.31	0.76	2.11			
RePI	Cl	98	0.043	0.22	−0.47	0.34	1	0.006	0.937
	C2	98	0.046	0.23	−0.57	0.52			
AbsPI	Cl	98	0.188	0.12	0.02	0.47	1	0.047	0.829
	C2	98	0.184	0.14	0.02	0.57			

PIB = PI baseline in middle cerebral artery (MCA); PIR = PI reactions immediately after defined auditory stimulation in MCA; RePI = relative PI changes; AbsPI = absolute PI changes.

**Table 2 ijms-21-03855-t002:** A partial correlation coefficient between AbsPI and PIR in subgroups C1−, C1+, C2−, and C2+.

Group	Correlation between AbsPI and PIR		
	Pearsons Correlation Coefficient r	Effect Size R^2^	Significance *p*
Cl−	0.592	0.350	0.001
C1+	−0.574	0.329	0.000
C2−	0.344	0.118	0.047
C2+	−0.448	0.201	0.000

AbsPI = absolute PI changes; PIR = PI reactions immediately after defined auditory stimulation in MCA.

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
