# Peer review of "Fetal Middle Cerebral Artery Pulsatility Index in No-Risk Pregnancies: Effects of Auditory Stimulation and Pregnancy Order"

_ijms, 2020, doi:10.3390/ijms21113855_

Round 1

Reviewer 1 Report

Thank you for the opportunity to read this manuscript. Understanding prenatal hearing maturation and establishing reliable testing is an important topic and this groups seems to have published data on this aspect already. Intrauterine reaction to sound and later psychosocial development is an interesting area, however the authors discuss this (reaction type) but do not mention an ongoing follow-up of this cohort.

The introduction is well written. The sentence in line 72-74 is not easy to understand. What do the authors mean by % of responses (in one individual or within the group of individuals?)? The sentence needs to be revised.

Material and methods: The selection process is not clear, who was selected and why, was there a random selection? Which database do the authors talk about? What was the matching process, how was this done and why? When did this study take place? Is this the same cohort that has been used in previous publications of the study group or was this a new group?

Results: I would suggest like a more detailed description of the study group. Maternal and gestational age are given with 5 decimal points…. However, I miss information about fetal weight, the rate of IUGR infants, maternal blood pressure, gestational diabetes and a proxy for social class (e.g. maternal education). The table format should be changed to improve readability. Why do the authors give the mean of the whole group if they are only interested in the differences between the two groups? Figure 1: What d the lines show? I expect the red line symbolises the 50th centile, but what are the other 2 lines 10th and 90th or 3rd and 97th? Line 101-102 I do not understand why the values are under the normal reference range when most are within and only a few are above the shown curve. Line 111 please give p-values or levels of significance in table A1. I do not understand why table A2 is needed. Line 133-140: why should two values in the same individual measured within a very short time frame not be correlated?

Discussion should be more focused. Why discuss absolute values of PIB und PIR in such detail if the message is that the difference between whose two measures should be used to assess fetal hearing response.

Reviewer 2 Report

The reviewed work raises an interesting and important topic. However, the work has many shortcomings that significantly affect the quality of the manuscript.

Introduction  -

The introduction is based for the most part on a bibliography from 20 years ago, only two items come from the last 5 years.

Results -

Table A1 and A2 are illegible and unnecessarily extensive. They should be combined into one transparent table.

I have doubts about the correctness of the statistical methods used - why the ANOVA test was used to compare the two research groups.

Figure A1 is difficult to interpret - no proper descriptions.

Table A3 is unnecessary - the information contained therein is not relevant and can be presented in the text.

Figure A2 - no axis signatures.

The presented results are incorrectly presented, they do not lead to constructive conclusions, e.g. line 152-154 or are obvious, e.g. line 145-147.

Discussion -

Due to incorrectly presented results, the review of the discussion does not make sense.

Materials and Methods –

There is no information on the statistical methods used.

Round 2

Reviewer 1 Report

Overall, the manuscript has improved significantly.

I have no further comments on the introduction.

Methods:

Even though the study design is described now, I cannot decide from the presented data whether the data are representative of all pregnant women in the Serbian population. The authors do not present the response rate of the overall study. How many were asked, how many did not participate (with reasons), how many were excluded? They do not present any socio-economic variables of those not-participating, those 1000 participants and those 198 finally included in this analysis.

The authors mention a linear regression analysis (L759), however, I did not find this in the result section.

Results:

L323 is a discussion not a result, while L422  is a result not described in the methods.

Figure A1: why not label the y-axis as PIB in line with the wording in the rest of the manuscript?

Figure A2: is this figure really necessary? Those two measures should be correlated or the whole method would not make sense.

Figure A3: why do the authors give just the mean of pooled data of each week of gestation? Why not present data for all?

Discussion:

Is there no reference for these equations? This line of arguing is going far away from the rest of the manuscript.  Even though it might be interesting, it distracts from the simple message of the manuscript which parameter should be used to assess prenatal hearing maturation in both nulliparous and multiparous women.

Reviewer 2 Report

Although the authors responded to most of my suggestions, however, I still notice some shortcomings Figure A3 (formerly A2) - the authors present the same relationships as in the previous version, but the calculations for this figure are different than before - is this the result of an error or have the groups been changed? Manuscrypt has many punctuation errors.
